# Data symmetries generate drifting similarity matrices in manifold-tiling neural codes

**Farhad Pashakhanloo**[1] and **Jacob A. Zavatone-Veth**[1,2]
[1]Center for Brain Science and [2]Society of Fellows
Harvard University, Cambridge, MA, USA
{fpasha,jzavatoneveth}@fas.harvard.edu

**Editors:** Marco Fumero, Clementine Domine, Zorah Lähner, Irene Cannistraci, Bo Zhao, Alex Williams

## Abstract

What can representational similarity matrices tell us about a neural code? As the popularity of these summary statistics grows, so too does the need for a complete characterization of their properties. Here, we study how functionally-irrelevant degrees of freedom affect representational similarity matrices in perhaps the simplest nonlinear neural code: one with localized receptive fields tiling a symmetric manifold. Stimulus symmetries render many tilings functionally equivalent, but these configurations yield different similarity matrices provided that the tiling is sparse. We show that stochastic gradient descent or energetic regularization can generate sparse, drifting tilings, leading in turn to drifting similarity matrices. Our results illustrate the challenges inherent in comparing non-linear neural codes, when functionally-equivalent representations are not related by a simple rotation.

## 1 Introduction

The brain represents many aspects of the sensory world through manifold-tiling neural representations, in which the neurons' receptive fields form a regular lattice over stimulus space [1, 2]. At the same time, the sensory world is replete with symmetries. This leads to an ambiguity: given a particular spacing between receptive fields, there exist many possible tilings of the stimulus manifold which are functionally equivalent given the symmetry. Therefore, to completely specify the representation, one must fix a gauge; one must anchor the lattice of receptive fields.[1]

Here, we consider the consequences of this symmetry for common approaches to quantifying neural representations based on representational similarity matrices (RSMs) [4–9]. We first show that the RSM in a toy model of neurons with receptive fields tiling a circle is dependent on the global orientation of the arrangement of receptive fields. However, this dependence rapidly diminishes as the number of distinct receptive field centers increases, raising the possibility that this effect can be neglected in large neural populations. We show that this is not always the case: stochastic gradient descent (SGD) or energetic regularization can lead to solutions where the receptive fields of multiple neurons collapse together to a common central location. Critically, even after receptive fields collapse, SGD does not fix the gauge, and the RSM drifts over time. Varying RSMs are also generated upon training from different initial weight configurations.

---

[1]Our use of the term "gauge" is inspired by its use in physics, though the phenomenology we describe here is not identical, as in this case the gauge is manifestly observable though not functionally relevant [3].

Proceedings of the III edition of the Workshop on Unifying Representations in Neural Models (UniReps 2025).

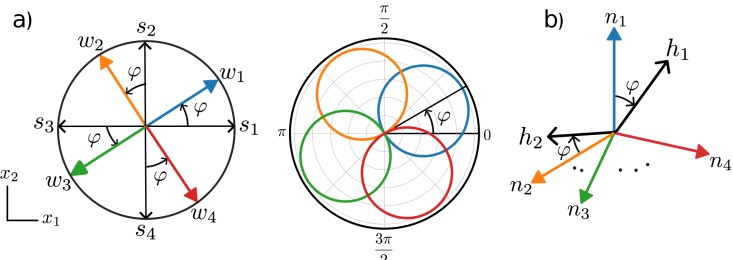

Figure 1: Neuron tuning and representations in a toy model. a) Left: RF center vectors ($w_i$) of four neurons tiling a one-dimensional ring. Right: corresponding angular tuning curves. b) Representations $h_1$ and $h_2$ of two stimuli $s_1$ and $s_2$ in the four-dimensional representation space.

## 2 Manifold-tiling solutions in a toy model

As a start, consider a simple toy setup where data lies on a one-dimensional ring ($x \in \mathbb{S}^1$), and $n$ ReLU neurons tile this space uniformly. The activation of $i$-th neuron in response to an input $x$ is

$$h_i(x) = \text{ReLU}(w_i^\top x), \quad \text{where} \quad w_i = (\cos\theta_i, \sin\theta_i) \tag{1}$$

is the (unit-length) weight vector associated with that neuron, defined by an angle

$$\theta_i = 2(i-1)\pi/n + \varphi \quad \text{for} \quad i \in [n]. \tag{2}$$

Here, $\varphi$ is an arbitrary offset angle and the only degree of freedom for this configuration (it is the gauge variable). Since each neuron has a receptive field (RF) that covers half the space ($\pi$ radians on the ring), $n = 4$ is the minimum number of neurons that can cover this space faithfully (Fig. 1a; see Appendix A). In this configuration, one can track the neuronal representations of a set of trial stimuli, $s_1 = (1,0)^\top, s_2 = (0,1)^\top, s_3 = (-1,0)^\top$ and $s_4 = (0,-1)^\top$, which for $\varphi \in [0, \pi/2]$ become:

$$h_1 = \begin{pmatrix} \cos(\varphi) \\ 0 \\ 0 \\ \sin(\varphi) \end{pmatrix}, \quad h_2 = \begin{pmatrix} \sin(\varphi) \\ \cos(\varphi) \\ 0 \\ 0 \end{pmatrix}, \quad h_3 = \begin{pmatrix} 0 \\ \sin(\varphi) \\ \cos(\varphi) \\ 0 \end{pmatrix}, \quad h_4 = \begin{pmatrix} 0 \\ 0 \\ \sin(\varphi) \\ \cos(\varphi) \end{pmatrix}. \tag{3}$$

Increasing $\varphi$ beyond $\pi/2$ is equivalent to circularly permuting the neuron labels and taking the residue of $\varphi$ modulo $\pi/2$. If $H = [h_1, h_2, h_3, h_4]$ is a matrix that contains all the representations of trial stimuli, its corresponding RSM is:

$$RSM = H^\top H = \begin{pmatrix} 1 & \rho(\varphi) & 0 & \rho(\varphi) \\ \rho(\varphi) & 1 & \rho(\varphi) & 0 \\ 0 & \rho(\varphi) & 1 & \rho(\varphi) \\ \rho(\varphi) & 0 & \rho(\varphi) & 1 \end{pmatrix}, \quad \text{where} \quad \rho(\varphi) = \sin\varphi\cos\varphi \tag{4}$$

should be evaluated using the residue of $\varphi$ modulo $\pi/2$ (see Appendix A). The above suggests that even in such a simple case where the gauge variable $\varphi$ corresponds to an overall rotation of the tuning curves, the RSM depends on it. This is because the global rotation of the tuning curves does not translate to an overall rotation in the representation space. This is illustrated geometrically in Fig. 1b, where the angle between the representations of two stimuli can change, yielding a variable RSM.

### 2.1 The RSM becomes asymptotically gauge-invariant when the tiling is dense

The $n = 4$ case above corresponded to the minimum number of ReLU neurons needed to tile the ring. Here, we study how this effect changes as the number of neurons increases. In Appendix A, we show that for $n$ a multiple of 4 the RSM has a simple closed-form expression which generalizes that found above for $n = 4$; all that changes is the constant value of the diagonal elements and the details of the function $\rho(\varphi)$. We quantify the $\varphi$-dependence of the RSM by its normalized range:

$$\Delta_n := \frac{\max_\varphi(RSM_{off}) - \min_\varphi(RSM_{off})}{RSM_{diag}},$$

where $RSM_{off}$ and $RSM_{diag}$ are the unique non-zero off-diagonal and diagonal parts of the RSM, respectively.

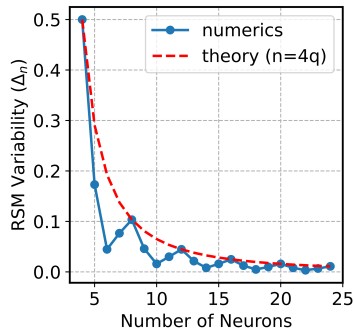

Figure 2: RSM variability as a function of number of neurons.

This is plotted in Figure 2 alongside the theoretical prediction

$$\Delta_n = \frac{2}{n} \tan\left(\frac{\pi}{n}\right), \tag{5}$$

which holds for $n$ a multiple of 4. As $n \to \infty$, the variability in the RSM vanishes as $\mathcal{O}(n^{-2})$.

## 3 Learning-induced manifold-tiling and drift

So far, we showed that a manifold-tiling solution can create a varying RSM as a function of the functionally-irrelevant gauge variable, provided that the tiling is not too dense. Here, we study this in a learning scenario. This includes a two-layer autoencoder with ReLU neurons and trained under SGD and weight decay. We give more details and additional experimental results in Appendix C.

### 3.1 SGD does not favor solutions with a particular RSM

As shown in Fig. 3, SGD training replicates the uniform-tiling solution for the $n = 4$ neurons in the hidden layer. Importantly, over the course of continual training, the tuning curves and the RSM drift over time without a clear preference for any particular solution. This ambivalence also holds across different realizations of the learning started from different initializations of the weights (see Fig S1).

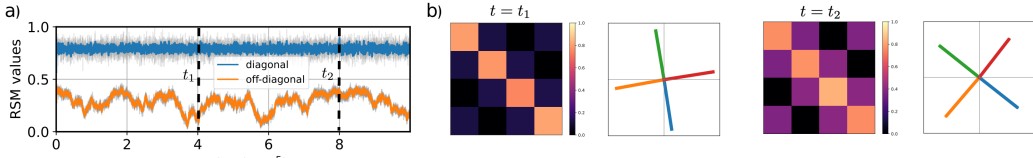

Figure 3: Continual training under SGD leads to a variable RSM in a ReLU network with two-dimensional input and $n = 4$ neurons. a) Values of RSM over time. b) At two time snapshots of training, the RSM matrices (top), and the corresponding weight vectors (bottom) are shown.

### 3.2 SGD inductive bias or energetic regularization can generate sparsely-tiling solutions

We found in Section 2.1 that having a large number of neurons that densely tile the manifold suppresses variability in the RSM. Here, we demonstrate two ways that learning can lead to solutions with highly-variable RSMs, even when the number of hidden neurons is large. First, as shown in Figure 4, SGD can lead the receptive fields of multiple neurons to collapse (align) together, leading to a solution that *effectively* only has four neurons. We observed that this collapse can occur when SGD noise is large (see Fig. S2 for an example of a low-noise regime). Second, regularizing the activations with an $L_1$ penalty—to mimic energetic costs—can lead to all but four neurons becoming silent (see Fig. 5). These effects can allow RSM variability to persist.

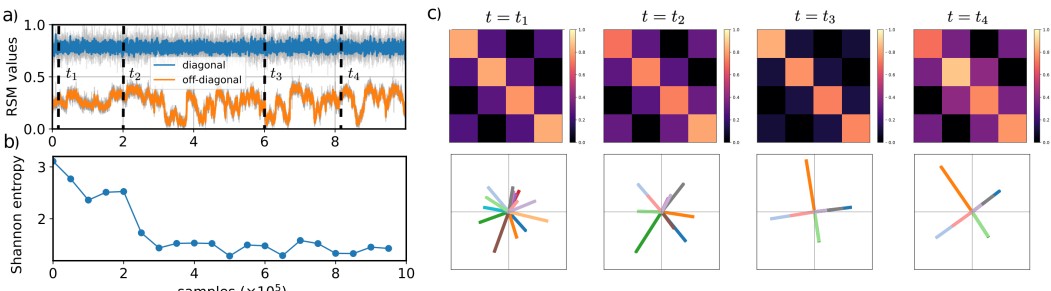

Figure 4: Collapse of neurons under SGD training for two-dimensional input and $n = 15$ neurons. a) Values of RSM over time. b) Entropy of the distribution of cosine of pairwise angles between all neurons. c) RSMs (top), and the corresponding weight vectors (bottom) at different snapshots during training. Alignment of neurons' weights into 4 orthogonal directions is evident through learning.

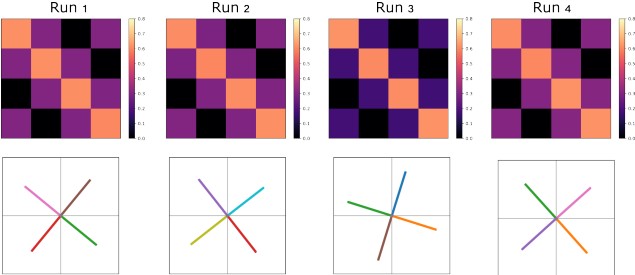

Figure 5: RSM plots (top) and neurons' weight vectors (bottom) for four different runs under $l_1$ penalty on activations ($n = 15$ neurons). Similar to Figure S2, the batch size is large but the $l_1$ penalty on the hidden-layer activations leads to all but 4 neurons remaining active. Each simulation is run with $5 \times 10^4$ samples seen (batch size of 100).

## 3.3 Structured RSM variability for higher-dimensional stimuli

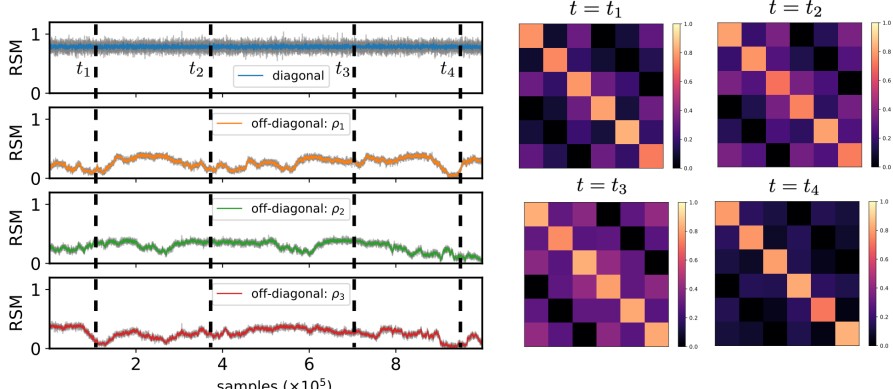

Figure 6: RSM variability as a result of continual training for three-dimensional input and $n = 6$ neurons. (Left) Components of the empirical RSM as a function of time. The non-zero off-diagonal elements are grouped into three distinct sets ($\rho_1$, $\rho_2$, and $\rho_3$) based on the predictions derived in Appendix B (gray curves correspond to the group members and the colored curves denote the group averages). (Right) RSM matrices at different snapshots demonstrate variability through time.

We have thus far restricted our attention to a two-dimensional stimulus with a one-dimensional symmetry group (rotations in the plane). However, the same gauge redundancy is present in neural codes tiling higher-dimensional symmetric manifolds. Analytical study of tiled representations of higher-dimensional manifolds is challenging because the maximally uniform arrangement of receptive field centers is unknown in general, even for the sphere $\mathbb{S}^2$ [10]. However, if we tile $\mathbb{S}^{d-1}$ with a reflection-symmetric arrangement of $2d$ neurons—so that each neuron is paired with another that has an oppositely-oriented receptive field—we can predict the structure of variability in the RSM induced by the $d(d-1)/2$-dimensional $\mathcal{O}(d)$ symmetry (see Appendix B). In general, the RSM is given by

$$RSM = \begin{pmatrix} I_d + R & R \\ R & I_d + R \end{pmatrix}, \tag{6}$$

where $R$ is a $d \times d$ symmetric matrix with zeros along the diagonal, which thus has $d(d-1)/2$ independent elements. In three dimensions, there are therefore three sets of distinct non-zero off-diagonal elements of the RSM,

$$R = \begin{pmatrix} 0 & \rho_1 & \rho_2 \\ \rho_1 & 0 & \rho_3 \\ \rho_2 & \rho_3 & 0 \end{pmatrix}, \tag{7}$$

each of which varies over time during continual training (Fig. 6). Not only is the RSM highly structured in the sense that its elements cluster into groups, but the variability of these groups is correlated (see also Fig. S3 for $d = 10$, where there are $d(d-1)/2 = 45$ distinct groups).

# 4 Discussion

Representational similarity measures have been used extensively in neuroscience and ML to compare internal representations of artificial and biological networks [4, 8, 9, 11]. A body of theoretical work on the statistical physics of large ensembles of neurons suggest that various summary statistics—including RSMs—may be stable when the number of neurons in each hidden layer of the network tend to infinity [9]. Indeed, our result in Section 2.1 is compatible with those results, as the variability of RSM asymptotically approaches zero when the tiling is dense. However, we showed that there are regimes in which learning may lead to solutions with a low *effective* number of neurons. In evaluating how high the effective neuron count must be in order to suppress variability in the RSM, one must of course consider the intrinsic dimensionality of the symmetric stimulus manifold. By the curse of dimensionality, we expect that the number of neurons must scale exponentially with intrinsic dimension to achieve comparably low variability (see Appendix B).

Ongoing drift in representations despite stable task performance has been observed in the brain and in artificial neural networks [12, 13]. Such changes, often associated with symmetries in the parameter space of the model, may nonetheless lead to stable representational similarity matrices [13–16]. However, there is also experimental evidence showing changing similarity measures over time in cortex [13, 17]. Here, we show that, in the presence of nonlinearity, simple transformations of symmetric inputs could lead to nonlinear deformation of the representations that in turn lead to drifting RSMs. This is especially striking given that the type of solution is not changing. In this regard, it is also in contrast to the linear case studied in [18], where for the minimum-norm solutions the RSM was found to be unique. In more general regularized autoencoders, recent works have begun to disentangle the conditions under which unique axis-aligned solutions are preferred [19–21]. However, how symmetries in data affect these results remains unknown.

The underlying reason why the RSM is not gauge-invariant is that solutions with different gauge angle are not related by a simple rotation in representation space. To compare representations in a way that is invariant to this symmetry, one must design a metric that is invariant to these transformations. Even though this may require knowledge of the symmetries of the stimulus space—which in many realistic scenarios are unknown—our study motivates such attempts in the future.

## Acknowledgments and Disclosure of Funding

We thank Juan Carlos Fernández del Castillo and William Qian for helpful comments on a previous version of this manuscript. F.P. was supported by the Harvard Center for Brain Science (CBS)-NTT Fellowship Program on the Physics of Intelligence. J.A.Z.-V. and this research were supported by the Office of the Director of the National Institutes of Health under Award Number DP5OD037354, and additionally by a Junior Fellowship from the Harvard Society of Fellows.

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

## Appendices

## A  The toy orientation tuning model with many neurons

In the main text, we focused on the case where the toy orientation tuning model has only $n = 4$ neurons. This is the minimum number of neurons required to faithfully reconstruct the ring because (1) at least one neuron must respond to every stimulus, and (2) if only a single neuron responds to some set of stimuli, then responses may be ambiguous due to the symmetry in each neuron's response under reflection about the axis of its preferred stimulus. This, for instance, means that $n = 3$ neurons does not suffice, as with equidistributed receptive field centers there are thus regions around each neuron's receptive field center where only a single neuron is active. In this Appendix, we consider $n > 4$ neurons. We show that the RSM for the toy orientation tuning model has a simple form whenever the number of neurons is an integer multiple of four.

As in the main text, the activation of the $i$-th of $n$ neurons is given by $h_i(x) = \text{ReLU}(w_i^\top x)$, where $w_i = (\cos\theta_i, \sin\theta_i)^\top$ for $\theta_i = 2(i-1)\pi/n + \varphi$. The four test stimuli of interest are $s_1 = (1,0)^\top$, $s_2 = (0,1)^\top$, $s_3 = (-1,0)^\top$, and $s_4 = (0,-1)^\top$, and the RSM has elements $RSM_{ab} = \sum_{i=1}^n h_i(s_a)h_i(s_b)$ for $a, b \in [4]$.

By symmetry, all diagonal elements of the RSM are equal to:

$$RSM_{diag} = \sum_{k=0}^{n-1} \text{ReLU}\left(\cos\left(\frac{2\pi k}{n} + \varphi\right)\right)^2,\tag{A.1}$$

and non-zero off-diagonal elements are all equal to:

$$\rho = \sum_{k=0}^{n-1} \text{ReLU}\left(\cos\left(\frac{2\pi k}{n} + \varphi\right)\right)\text{ReLU}\left(\sin\left(\frac{2\pi k}{n} + \varphi\right)\right).\tag{A.2}$$

We can see that increasing $\varphi$ by integer multiples of $2\pi/n$ is equivalent to shifting the index $k$, which has the effect of circularly permuting neuron labels. As the RSM is invariant under permutation of the neuron labels, it must therefore be periodic in $\varphi$, with period $2\pi/n$. Therefore, it suffices to consider $\varphi \in [0, 2\pi/n)$. We therefore put $\varphi = 2\pi\delta/n$ for $\delta \in [0, 1)$.

### A.1  Evaluation of the similarity matrix elements

First, consider the off-diagonal elements. As $\cos(\theta)$ and $\sin(\theta)$ are simultaneously positive only for $\theta \in [0, \pi/2)$, this is equivalent to

$$\rho = \sum_{k=0}^{n-1} \cos\left(\frac{2\pi(k+\delta)}{n}\right) \sin\left(\frac{2\pi(k+\delta)}{n}\right) \mathbf{1}_{0 \leq k+\delta < n/4}\tag{A.3}$$

Supposing that $n = 4q$ for an integer $q \geq 1$, we see that $0 \leq k + \delta < n/4$ only if $k \leq q - 1$, for any $\delta \in [0, 1)$. Therefore,

$$\rho = \sum_{k=0}^{q-1} \cos\left(\frac{\pi(k+\delta)}{2q}\right) \sin\left(\frac{\pi(k+\delta)}{2q}\right)\tag{A.4}$$

This is at last a sum that can be evaluated, giving

$$\rho = \frac{1}{2}\cot\left(\frac{\pi}{2q}\right)\cos\left(\frac{\pi\delta}{q}\right) + \frac{1}{2}\sin\left(\frac{\pi\delta}{q}\right)\tag{A.5}$$

or, in terms of $n$ and $\varphi$,

$$\rho = \frac{1}{2}\cot\left(\frac{2\pi}{n}\right)\cos(2\varphi) + \frac{1}{2}\sin(2\varphi).\tag{A.6}$$

As $\cos(\theta)$ is positive if $\theta \in [0, \pi/2) \cup (3\pi/2, 2\pi)$, we can similarly evaluate the diagonal elements as

$$\sum_{k=0}^{n-1} \cos\left(\frac{2\pi(k+\delta)}{n}\right)^2 \left[\mathbf{1}_{0 \leq k+\delta < q} + \mathbf{1}_{3q < k+\delta < 4q}\right] \tag{A.7}$$

$$= \sum_{k=0}^{q-1} \cos\left(\frac{\pi(k+\delta)}{2q}\right)^2 + \sum_{k=3q}^{4q-1} \cos\left(\frac{\pi(k+\delta)}{2q}\right)^2 \tag{A.8}$$

$$= q. \tag{A.9}$$

Thus, letting

$$\rho_n(\varphi) = \frac{1}{2}\cot\left(\frac{2\pi}{n}\right)\cos(2\varphi) + \frac{1}{2}\sin(2\varphi), \tag{A.10}$$

we find that for any $n$ a multiple of four we have

$$RSM = \begin{pmatrix} \frac{n}{4} & \rho_n(\varphi) & 0 & \rho_n(\varphi) \\ \rho_n(\varphi) & \frac{n}{4} & \rho_n(\varphi) & 0 \\ 0 & \rho_n(\varphi) & \frac{n}{4} & \rho_n(\varphi) \\ \rho_n(\varphi) & 0 & \rho_n(\varphi) & \frac{n}{4} \end{pmatrix} \tag{A.11}$$

for any $\varphi \in [0, 2\pi/n)$. For larger $\varphi$, we can apply this result using $\varphi \mod 2\pi/n$.

We can evaluate by a similar argument the population-averaged firing rate for each of the test stimuli:

$$\bar{r} = \frac{1}{n} \sum_{k=0}^{n-1} \cos\left(\frac{2\pi(k+\delta)}{n}\right) \left[\mathbf{1}_{0 \leq k+\delta < q} + \mathbf{1}_{3q < k+\delta < 4q}\right] \tag{A.12}$$

$$= \frac{1}{n} \sum_{k=0}^{q-1} \cos\left(\frac{\pi(k+\delta)}{2q}\right) + \frac{1}{n} \sum_{k=3q}^{4q-1} \cos\left(\frac{\pi(k+\delta)}{2q}\right) \tag{A.13}$$

$$= \frac{1}{n} \csc\left(\frac{\pi}{4q}\right) \cos\left(\frac{\pi - 2\pi\delta}{4q}\right) \tag{A.14}$$

$$= \frac{1}{n} \csc\left(\frac{\pi}{n}\right) \cos\left(\frac{\pi}{n} - \varphi\right), \tag{A.15}$$

where again in the last line we consider $\varphi \in [0, 2\pi/n)$.

## A.2 Large-$n$ limit

We now want to study what happens when we take the number of neurons to be large. To do so, it is convenient to use the parameterization of the mean firing rate and RSM in terms of $q$ and $\delta$, as the period of these objects in $\delta$ is defined to be independent of $n$. First, the mean firing rate expands as

$$\bar{r} = \frac{1}{4q} \csc\left(\frac{\pi}{4q}\right) \cos\left(\frac{\pi - 2\pi\delta}{4q}\right) \tag{A.16}$$

$$= \frac{1}{\pi} - \frac{\pi(1 - 6\delta + 6\delta^2)}{48q^2} + \mathcal{O}\left(\frac{1}{q^4}\right). \tag{A.17}$$

The normalized non-zero off-diagonal elements of the RSM are given by

$$\frac{1}{2q}\cot\left(\frac{\pi}{2q}\right)\cos\left(\frac{\pi\delta}{q}\right) + \frac{1}{2q}\sin\left(\frac{\pi\delta}{q}\right) = \frac{1}{\pi} - \frac{\pi(1 - 6\delta + 6\delta^2)}{12q^2} + \mathcal{O}\left(\frac{1}{q^4}\right). \tag{A.18}$$

This shows that the mean firing rate and RSM are gauge-invariant only asymptotically, in the limit $n \to \infty$.

## A.3 Normalized variability

As in the main text, we consider the normalized variability

$$\Delta_n = \frac{\max_\varphi RSM_{off} - \min_\varphi RSM_{off}}{RSM_{diag}}. \tag{A.19}$$

Using the expression for $RSM_{off}$ from above, we see that it is maximized at $\delta = 1/2$, where it takes value

$$\frac{1}{2}\cot\left(\frac{\pi}{2q}\right)\cos\left(\frac{\pi\delta}{q}\right) + \frac{1}{2}\sin\left(\frac{\pi\delta}{q}\right)\bigg|_{\delta=1/2} = \frac{1}{2}\csc\left(\frac{\pi}{2q}\right) \tag{A.20}$$

and minimized at $\delta = 0$, where it takes value

$$\frac{1}{2}\cot\left(\frac{\pi}{2q}\right)\cos\left(\frac{\pi\delta}{q}\right) + \frac{1}{2}\sin\left(\frac{\pi\delta}{q}\right)\bigg|_{\delta=0} = \frac{1}{2}\cot\left(\frac{\pi}{2q}\right). \tag{A.21}$$

Using the identity $\csc(\theta) - \cot(\theta) = \tan(\theta/2)$, we thus have that

$$\Delta_{4q} = \frac{1}{2q}\tan\left(\frac{\pi}{4q}\right), \tag{A.22}$$

or

$$\Delta_n = \frac{2}{n}\tan\left(\frac{\pi}{n}\right) \tag{A.23}$$

in terms of $n$. As $\tan(\theta) = \theta + \mathcal{O}(\theta^3)$ as $\theta \downarrow 0$, we thus find that

$$\Delta_{4q} = \frac{\pi}{8q^2} + \mathcal{O}\left(\frac{1}{q^4}\right) \tag{A.24}$$

as $q \to \infty$, which is of course equivalent to

$$\Delta_n = \frac{2\pi}{n^2} + \mathcal{O}\left(\frac{1}{n^4}\right). \tag{A.25}$$

## A.4 Penalizing the $L_1$ norm of activations does not fix a gauge

In the main text, we showed that one way of promoting RSM variability with a large number of neurons is the presence of energetic cost (such as the L1-penalty) on the activations. Here, we show that when data is uniformly distributed on the manifold, L1-penalty does not fix a gauge. Observe that the response of the $i$-th neuron to a stimulus $x = (\cos\psi, \sin\psi)^\top$ is simply

$$h_i(x) = \text{ReLU}\left(\cos\left(\frac{2\pi(i-1)}{n} + \varphi - \psi\right)\right), \quad i \in [n]. \tag{A.26}$$

Then, the $L_p$ norm of the hidden layer activations, assuming a uniform angular distribution, is

$$\|h\|_p = \left(\int_0^{2\pi}\frac{d\psi}{2\pi}\bar{r}_{n,p}(\varphi - \psi)\right)^{1/p}, \tag{A.27}$$

where we let

$$\bar{r}_{n,p}(\varphi - \psi) = \sum_{k=0}^{n-1}\text{ReLU}\left(\cos\left(\frac{2\pi k}{n} + \varphi - \psi\right)\right)^p. \tag{A.28}$$

Though we have assumed that everything is normalized, we remark that if we allowed radial variation in the stimulus or in the weight vector, we could ignore it in this analysis. This is because the positive-homogeneity of the ReLU means that adding a variable radius will simply multiply the formula above by a constant independent of both $n$ and $\varphi$.

By an argument identical to our previous analysis, $\bar{r}_{n,p}(\varphi - \psi)$ must be $2\pi/n$-periodic in its argument. Therefore, by averaging over $\psi \in [0, 2\pi)$ we are averaging over $n$ periods, which in turn means that $\|h\|_1$ must be independent of the gauge angle $\varphi$.

How do deviations of the stimulus distribution from perfect uniformity affect this result? Recalling our results from before, we have for $n$ an integer multiple of four that

$$\bar{r}_{n,1}(\varphi - \psi) = \csc\left(\frac{\pi}{n}\right)\cos\left(\frac{\pi}{n} - (\varphi - \psi)\right), \tag{A.29}$$

where we should plug in the residue of $\varphi - \psi$ modulo $2\pi/n$, while

$$\bar{r}_{n,2}(\varphi - \psi) = \frac{n}{4}. \tag{A.30}$$

This means that for $n$ a multiple of 4 penalizing the $L_2$ norm of activations cannot fix a gauge for the trivial reason that it is stimulus-independent. In contrast, the $L_1$ norm is sensitive to fluctuations in the distribution of stimuli.

# B    Reflection-symmetric tilings of the sphere in higher dimensions

Performing a similarly-detailed analysis of the RSMs for tiling representations of higher-dimensional manifolds is challenging. Even for the ordinary sphere $\mathbb{S}^2$, explicitly writing down a tiling solution for an arbitrary number of neurons is challenging—this corresponds to a variant of the classic Thompson problem in potential theory, to which the general solution remains unknown [10]. On general grounds, we expect that the number of neurons required to suppress variability in the RSM should scale exponentially with the intrinsic dimension of the manifold. This intuition follows from the curse of dimensionality: assuming manifold-tiling localized receptive fields, the density of field centers must be high enough to compensate for the decay in overlap due to a change in gauge.

Though we cannot write down the RSM explicitly, we can characterize its overall structure for solutions with an even number of neurons tiling the sphere $\mathbb{S}^{d-1}$ in $d$ dimensions that obey a reflection-symmetry condition on the weights: for each neuron, there is another neuron with its receptive field oriented exactly opposite to the first. This corresponds to assuming that the stimulus-by-neuron weight matrix $W \in \mathbb{R}^{d \times 2n}$ has the form

$$W = (U^\top, -U^\top) \tag{B.1}$$

for some matrix $U \in \mathbb{R}^{n \times d}$.[2] Generalizing our study of the $d = 2$ case, we take our probe stimuli to be the standard basis vectors and their negations, *i.e.*, we have a stimulus matrix[3]

$$S = (I_d, -I_d) \in \mathbb{R}^{d \times 2d}, \tag{B.2}$$

whose representation is

$$H = \mathrm{ReLU}(W^\top S) = \begin{pmatrix} \mathrm{ReLU}(U) & \mathrm{ReLU}(-U) \\ \mathrm{ReLU}(-U) & \mathrm{ReLU}(U) \end{pmatrix}. \tag{B.3}$$

We show below that, assuming weight tying, demanding that the probe stimuli are faithfully reconstructed implies that the matrix $U$ must be semi-orthogonal, *i.e.*,

$$U^\top U = I_d. \tag{B.4}$$

If this holds, then the RSM has the general form

$$RSM = \begin{pmatrix} I_d + R & R \\ R & I_d + R \end{pmatrix} \tag{B.5}$$

where $R$ is a $d \times d$ symmetric matrix with zeros along the diagonal.

The main salient feature of this matrix for $d > 2$ is the fact that its elements obey non-trivial equality relations (relative to the case of the ring $\mathbb{S}^1$, where all non-zero off-diagonal elements of the RSM were equal). For instance, if $d = 3$, this yields an RSM of the form

$$RSM = H^\top H = \begin{pmatrix} 1 & \rho_1 & \rho_2 & 0 & \rho_1 & \rho_2 \\ \rho_1 & 1 & \rho_3 & \rho_1 & 0 & \rho_3 \\ \rho_2 & \rho_3 & 1 & \rho_2 & \rho_3 & 0 \\ 0 & \rho_1 & \rho_2 & 1 & \rho_1 & \rho_2 \\ \rho_1 & 0 & \rho_3 & \rho_1 & 1 & \rho_3 \\ \rho_2 & \rho_3 & 0 & \rho_2 & \rho_3 & 1 \end{pmatrix}. \tag{B.6}$$

where $\rho_1$, $\rho_2$, and $\rho_3$ are the three distinct non-zero elements of the matrix $R$. Here, the three functions $\rho_1$, $\rho_2$, and $\rho_3$ are continuous piecewise functions of the three gauge angles that appear in $d = 3$, which can be computed explicitly using MATHEMATICA—though their particular form is not illuminating. We see in Figure 6 that the equality relations between different elements of the RSM implied by (B.6) are in fact obeyed to high accuracy in experiment. In Figure S3 we show that we observe the corresponding generalized structure (B.5) empirically in SGD-trained networks for $d = 10$.

---

[2]Our results extend to larger networks where neurons are duplicated so long as the weights are appropriately normalized. We leave to future work a full investigation of when this form of the RSM applies to larger networks that are equivalent to these small networks thanks to further internal symmetries of the architecture [22, 23].

[3]As long as we assume that the test stimuli are given by a set of $d$ orthonormal vectors along with their negations, we lose no generality in making this choice because the global rotation symmetry implies that we can rotate the basis so that the test stimuli are axis-aligned.

## B.1 Structure of the reconstruction

We first show that the matrix $U$ must be semi-orthogonal in order to faithfully reconstruct the probe stimuli $S$. If we assume weight-tying, we then have the reconstruction

$$\hat{S} = W \operatorname{ReLU}(W^\top S) = (A, -A) \tag{B.7}$$

where

$$A = U^\top \operatorname{ReLU}(U) - U^\top \operatorname{ReLU}(-U) = U^\top \big[ \operatorname{ReLU}(U) - \operatorname{ReLU}(-U) \big]. \tag{B.8}$$

But, as $\operatorname{ReLU}(x) - \operatorname{ReLU}(-x) = x$ for any $x \in \mathbb{R}$, this simplifies to

$$A = U^\top U. \tag{B.9}$$

To exactly reconstruct the test stimuli, we should have $\hat{S} = S$ and thus

$$A = I_d, \tag{B.10}$$

which implies that the matrix $U$ must be semi-orthogonal:

$$U^\top U = I_d. \tag{B.11}$$

## B.2 Structure of the RSM

We now show that the RSM has the form (B.5). For this representation, we have the RSM

$$RSM = H^\top H = \begin{pmatrix} Q & R \\ R & Q \end{pmatrix} \tag{B.12}$$

where we have defined the $d \times d$ blocks

$$Q = \operatorname{ReLU}(U)^\top \operatorname{ReLU}(U) + \operatorname{ReLU}(-U)^\top \operatorname{ReLU}(-U) \tag{B.13}$$

$$R = \operatorname{ReLU}(U)^\top \operatorname{ReLU}(-U) + \operatorname{ReLU}(-U)^\top \operatorname{ReLU}(U), \tag{B.14}$$

Noting that $R$ is symmetric, to prove that the RSM takes the form (B.5) it suffices to show that $R$ has zeros along the diagonal and that $Q = U^\top U + R$.

The fact that $R$ has zeros along the diagonal is easy to see upon expanding in indices:

$$R_{ii} = \sum_{k=1}^{n} [\operatorname{ReLU}(U_{ki}) \operatorname{ReLU}(-U_{ki}) + \operatorname{ReLU}(-U_{ki}) \operatorname{ReLU}(U_{ki})] = 0, \tag{B.15}$$

as $\operatorname{ReLU}(x) \operatorname{ReLU}(-x) = 0$ for all $x \in \mathbb{R}$.

To show that $Q = U^\top U + R$, we observe that for $x, y \in \mathbb{R}$ we have

$$\begin{aligned}
& \operatorname{ReLU}(x) \operatorname{ReLU}(y) + \operatorname{ReLU}(-x) \operatorname{ReLU}(-y) \\
& \quad - \operatorname{ReLU}(x) \operatorname{ReLU}(-y) - \operatorname{ReLU}(-x) \operatorname{ReLU}(y) \\
& = xy.
\end{aligned} \tag{B.16}$$

Applying this identity element-wise, we have

$$\begin{aligned}
Q - R &= \operatorname{ReLU}(U)^\top \operatorname{ReLU}(U) + \operatorname{ReLU}(-U)^\top \operatorname{ReLU}(-U) \\
& \quad - \operatorname{ReLU}(U)^\top \operatorname{ReLU}(-U) - \operatorname{ReLU}(-U)^\top \operatorname{ReLU}(U) \tag{B.17} \\
&= U^\top U. \tag{B.18}
\end{aligned}$$

This proves that $Q = U^\top U + R$, and thus that in general we have

$$RSM = \begin{pmatrix} U^\top U + R & R \\ R & U^\top U + R \end{pmatrix}. \tag{B.19}$$

Note that this applies for any set of weights obeying the reflection-symmetry condition. For a solution satisfying $U^\top U = I_d$, this reduces to (B.5). As $R$ is a $d \times d$ symmetric matrix with zeros along the diagonal, it has $d(d-1)/2$ independent elements, matching the dimensionality of the gauge group.

# C  Additional experimental details and results

## C.1  Experimental details

Here we provide experimental details regarding the simulations provided throughout the paper. Neural networks consisted of a two-layer autoencoder with input dimension $d$, hidden-layer with ReLU activation (no bias) consisting of $n$ neurons, and an output layer that was a reconstruction of the input ($y = x$). Networks were trained using a vanilla SGD with a fixed learning rate ($\eta$), a weight decay ($\gamma$) and varying batch size ($b$) – see below for the specifics of each simulations. Training was performed on PyTorch with NVIDIA GeForce RTX 2080 Ti GPU. Data were generated in an online way from normal Gaussian distribution in $\mathbb{R}^d$. Prior to the online training, a warm-up pretraining was performed with no weight decay and batch size of 128 (this corresponds to time 0 on the axes of the training plots). The plots of RSM values over time were created by first saving snapshots of the model at intervals of 100 time steps, and then applying a moving average filter with the window size of 5. The specific parameters for each figure are as follows:

Figure 3: $d = 2$, $n = 4$, $\eta = 0.1$, $\gamma = 0.1$, $b = 1$.

Figure 4: $d = 2$, $n = 15$, $\eta = 0.15$, $\gamma = 0.1$, $b = 1$.

Figure 5: $d = 2$, $n = 15$, $\eta = 0.1$, $\gamma = 0.1$, $b = 100$, $\lambda_1 = 0.001$ (L1-penalty coefficient).

Figure 6: $d = 3$, $n = 6$, $\eta = 0.1$, $\gamma = 0.1$, $b = 1$.

Figure S1: $d = 2$, $n = 4$, $\eta = 0.1$, $\gamma = 0.1$, $b = 1$.

Figure S2: $d = 2$, $n = 15$, $\eta = 0.1$, $\gamma = 0.1$, $b = 100$.

Figure S3: $d = 10$, $n = 20$, $\eta = 0.075$, $\gamma = 0.05$, $b = 1$.

## C.2 Additional figures

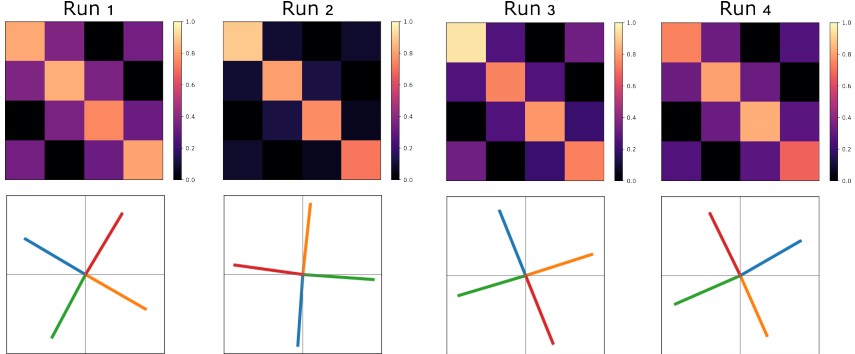

Figure S1: RSM plots (top) and neurons' weight vectors (bottom) for four different runs and $n = 4$ neurons. Each simulation is run with $5 \times 10^4$ samples seen (batch size of 1).

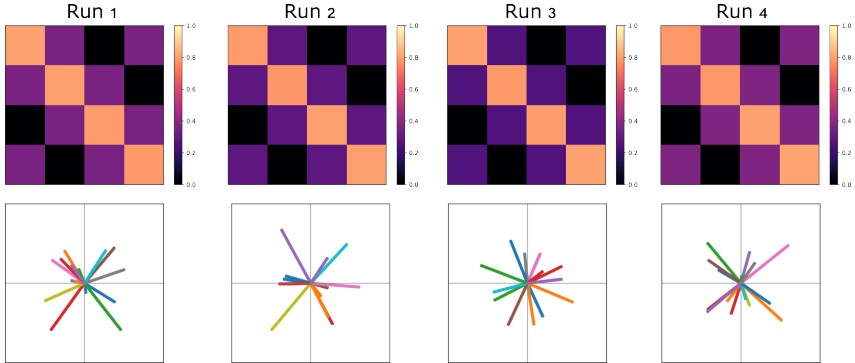

Figure S2: RSM plots (top) and neurons' weight vectors (bottom) for four different runs and $n = 15$ neurons. The batch size is large, leading to a relatively low SGD-noise regime where neurons' weights do not collapse. Each simulation is run with $5 \times 10^4$ samples seen (batch size of 100).

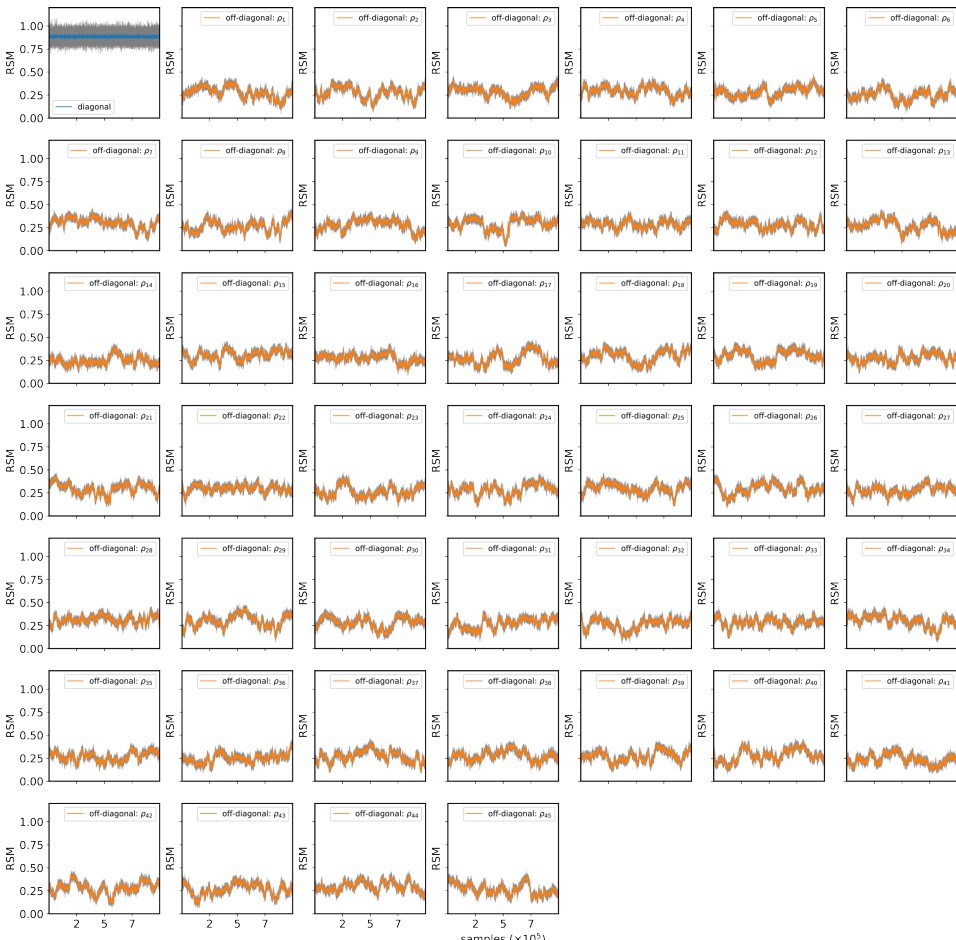

Figure S3: RSM values over time for a network with input dimension $d = 10$ and $n = 20$ neurons. Network is trained with SGD and batch size of one. Based on the predictions in Appendix B, the non-zero off-diagonal elements are placed into $d(d-1)/2 = 45$ groups. Each group contains 4 unique RSM values which, as predicted, are highly correlated (gray curves: members of the group, orange: group mean).