# OpenReview forum: "Data symmetries generate drifting similarity matrices in manifold-tiling neural codes"
_NeurIPS.cc/2025/Workshop/UniReps — UniReps2025_

### Official Review · Reviewer_KTQB · 2025-09-15

**Confidence:** 4

**Review:**

## Summary
This work presents a study of the variation in representational similarity matrices (RSM) when they are sparse at certain regimes, which defeats the purpose of RSM, which are used to compare internal representations of artificial and biological networks.

## Strengths
Good organization and development of ideas
Very detailed and clear
It provides clear and insightful ideas that can help increase the understanding of these representation methods.

## Weaknesses
Sometimes the jargon might make it difficult to follow when things can be explained in a simpler way; however, it is evident that it is well written.

**Score:**

4

**Topic Fit:**

3

---

### Official Review · Reviewer_ydo8 · 2025-09-15
**Conceptually strong, focused study of RSM drift**

**Confidence:** 3

**Review:**

### Summary
This paper shows that data symmetries can cause Representational Similarity Matrices (RSMs) to drift during training, even when task performance is stable. The authors introduce the concept of a functionally-equivalent "gauge" and show how SGD causes the network to drift between them, especially in sparse effective codes.

### Strength
- The paper is well-written with clear mathematical exposition and logical progression.
- The connection between data symmetries, gauge freedom, and RSM drift is novel and timely, and it is conceptually valuable for both neuroscience and ML.
- The toy model effectively builds intuition.

### Suggestions
- The analysis is confined to very simple architectures and idealized, symmetric data. Providing some preliminary evidence of drift on a real dataset (e.g., MNIST) or in deeper networks would be valuable.
- The observed collapse of neurons into sparse effective codes under high SGD noise is shown empirically. A more rigorous analysis of why noise drives this alignment, and how it depends on hyperparameters (e.g., learning rate, weight decay) would strengthen the contribution.

**Score:**

4

**Topic Fit:**

3

---

### Official Review · Reviewer_ZT6X · 2025-09-16
**This must be over my calibre...**

**Confidence:** 1

**Review:**

My best attempt at summarizing this paper is that it explores what conditions prevents/causes variability in representational similarity matrices (RSMs) for neural codes that tile a symmetric stimulus manifold.
I must admit that I am unsure what RSMs do/how they are used in the grand scheme of ML nor why variability of RSMs seem to be something that must be avoided. But it seems this is an important topic that highlights similarities between neuron representations in neuroscience and machine learning, which is on-topic for this workshop.

**Score:**

4

**Topic Fit:**

3